# Morphological Difficulties in People with Developmental Language Disorder

**DOI:** 10.3390/children9020125

**Published:** 2022-01-18

**Authors:** Esther Moraleda-Sepúlveda, Patricia López-Resa

**Affiliations:** Facultad de Ciencias de la Salud, Universidad de Castilla La Mancha, 45600 Talavera de la Reina, Spain; Patricia.lopezresa@uclm.es

**Keywords:** developmental language disorder, morphology, grammatical morphology

## Abstract

Background: One of the linguistic features of people with Developmental Language Disorder (DLD) is found in the use of different types of morphology. People with DLD have serious difficulties in the use of grammatical morphology, and some studies suggest that this factor could constitute a clinical marker of the disorder. The goal of this research was to discover the distinctive characteristics of the different morphological subareas in people with DLD. Methods: The sample consisted of 90 children and adolescents between 6 and 15 years old, of which 47 were diagnosed with DLD and the remaining 43 were in the Typical Development (TD) group. The two groups were matched in chronological age. The assessment instrument used was the morphology scale of the BLOC-C test, which includes 19 different subareas of morphology, among which are inflectional and verbal morphology. Results: The results have shown that people with DLD perform at a lower level than the TD group in all morphology subareas, as well as in the mean and percentile obtained. Conclusions: These data have led us to explore explanations based on several hypotheses and to confirm the research outlining the explicit morphological difficulties in people with DLD.

## 1. Introduction

Developmental Language Disorder, hereafter DLD, has been one of the most researched neurodevelopmental difficulties in recent years and is characterised by deficits in morphosyntactic constructions, lexical reduction and alterations in speech, social participation, communication and academic performance [1,2]. For this reason, it is essential to understand DLD as a heterogeneous disorder in its aetiology, manifestations and comorbidity. Along this line, although its estimated prevalence is around 7%—being slightly more common in men [3]—it is difficult to establish its prevalence with certainty, given the methodological differences used in the assessment and the diagnosis of DLD [4].

Regarding its characteristics, at a general level, research carried out in the Spanish language has revealed the negligible presence of communicative intention [5], lexical and grammatical errors [6], reduction in use of semantic resources [7], difficulties in understanding graphic humour [8] as well as communicative intentions [9]. People experiencing DLD also manifest problems in understanding figurative uses of language [7] and difficulties in understanding various grammatical structures [6].

Regarding the phonetic-phonological component, people with DLD experience multiple articulatory difficulties that often make their speech unintelligible [10]. Along the same lines, some studies report a reduction in syllabic structure and a limited syllabic configuration order, as well as a phonological inventory [1,11]. At the lexical level, recent studies have shown the presence of difficulties in the recognition of the oral lexicon, in social interaction and access to the lexicon [12].

As for the morphosyntactic component of people with DLD, it is characterised by the frequent omission and simplification of morphemes, prepositions, auxiliary verbs and pronouns, syntactic disorganisation, substitutions of the verb form and difficulties in distinguishing gender and number [13,14]. In addition, people with DLD have difficulties in sentence comprehension [15].

Specifically with regard to morphology, numerous studies have shown that people with DLD have difficulties in morphological acquisition, which has repercussions on their subsequent grammatical and linguistic development [16]. Thus, some research has reported that, although people with DLD have incorporated the notion of the plural, they cannot express the inflectional morphology of number on the verbal plane except by using the repetition of the name of the object [17,18]. That is, instead of forming a plural such as “apples”, people with DLD tend to use singular nouns, adding phrases such as “one more” or “another one”.

The inflectional verbal morphology in this population is also affected [1]. Along this line, the use of the verbal category is usually the most common impairment of the verbal inflectional morphology [19], manifesting itself through the frequent omission of verbal arguments, the use of simple verb forms from a semantic point of view, less verbal production, less diversity and complexity in verbal complementation and in the application of verbal inflectional marks in tense and aspect [20]. Other studies have observed a frequent juxtaposition in words as a result of the absence of relational elements or function words such as pronouns, determiners or prepositions [21].

Specifically, it appears that difficulties related to past tense morphology are particularly prominent in English-speaking children with DLD [22]. In this direction, Rice, Wexler and Cleave [22] reported that children with DLD produced a greater number of errors related to grammatical morphology. Some researchers have tried to explain these errors by hypothesising that DLD may involve a specific deficit when it comes to representing information about functional categories [23] and, consequently, in the application of certain grammatical rules [24]. However, subsequent research evidenced the productive capacity of children with DLD with inflected forms, thus giving rise to a new hypothesis focused on a period in which the inflected and unflexed roots are considered acceptable in the grammar of children with DLD [25]. For their part, Marchman, Saccuman and Wulfeck [23] suggested that the morphological acquisition pattern of children with DLD is similar to the pattern of their TD peers, it being posited that morphological errors tend to persist because children with DLD, unlike children with TD, continue operating with the system of learning rules. This hypothesis would explain the variability of the linguistic symptomatology at the morphological level of subjects with DLD. However, other studies put forward the possibility of a slowness or ineffectiveness of general processing mechanisms during the mediation of linguistic input [26].

Difficulties in morphology not only emerge when comparing children with DLD with others of the same chronological age, but also appear in children with Typical Development (TD) who are younger in age but matched in mean length of utterance (MLU) [22,27,28,29,30,31,32]. Therefore, it is evident that these difficulties may be a specific characteristic of this population.

These differences in the area of morphology have not only been evidenced in speakers of the Spanish or Castilian language, but these characteristics also appear in the same manner in other languages such as German [33], Swedish [34], Italian [29], French [35,36], Hebrew [37], Japanese [38,39] and Greek [40], among others. Therefore, evidence strongly suggests that this area constitutes a field of special differentiation.

The importance of the area of morphology resides in the fact that some investigations begin to establish these morphological difficulties in DLD as a possible clinical marker that distinguishes it from other language disorders [41,42].

So, the evidence shows that people with DLD experience specific difficulties in the area of morphology that manifest themselves when they develop syntactic structures [43]. Along this line, the goal of this study was to discover specifically what these specific morphological characteristics were in relation to the development of morphology in TD.

## 2. Materials and Methods

### 2.1. Participants

To carry out this study, a total of 90 children and adolescents of school age between 6 and 15 years (Primary Education and Secondary Education) participated, of which 47 were diagnosed with DLD and the remaining 43 with TD.

The groups were matched in chronological age. The characteristics of the sample are shown in Table 1. 

### 2.2. Instrument

In this research, the morphology section of the Objective and Criterial Language Battery (BLOC-C, by its Spanish acronym) by Puyuelo, Wiig, Renom and Solanas [44] was used as a tool to assess the language level of the participants. The battery evaluates four different components of language: morphology, syntax, semantics, and pragmatics. Its objective is to identify the areas in which the person has the most difficulty at a linguistic level. The evaluative experience has reflected its usefulness in populations with special educational needs [45]. In this work, given that the research is limited to the morphological component, the computerized version of the test was used to assess only the morphology section of the test. This area is made up of 19 subtests that evaluate different components: plural formation (with singular ending in consonant and vowel), adjectives, regular verb forms (present, past, future and imperfect), irregular verb forms (present, past and future), participles, comparatives, and superlatives, derived nouns (including professions), derived adjectives, personal pronouns (subject and object-based), possessive pronouns, and reflexive pronouns.

All the subtests contain 10 items to be evaluated, so the test offers a score in this area out of 190 points.

### 2.3. Procedure

The study was previously accepted by the respective bioethics committee of the Faculty of Health Sciences of the University of Castilla-La Mancha.

For data collection, the characteristics of the participants were considered. In the case of people with DLD, private clinics and DLD associations in Spain were contacted at the start of the study to inform them of the investigation and its characteristics. In this case, they were also asked to provide the reports where the diagnosis of DLD was analysed by a clinical specialist. In the case of people with TD, several schools were contacted to facilitate the sample. Subsequently, the schools that agreed to be part of this study, both in the case of DLD and in the case of people with TD, informed the parents/guardians of the research, respectively. All of them filled out an informed consent form prior to the assessment.

Once the consent was obtained, the BLOC was used in the assessment in order to obtain the study data. The estimated time during the assessment process with each child was a session of approximately 45 min for the morphology test.

## 3. Results

Given the normal distribution of the sample and to verify the study hypothesis (group with TD have better scores in the morphology subareas than group with DLD), we proceeded to choose the corresponding statistical analysis. To compare the results between both experimental groups, the Student’s t test of independent samples was used with the statistical program SPSS 24.0. This test is used to determine if there is a significant difference between the means of two groups. The results obtained are shown below in Table 2.

The data obtained show how all the scores in the different subareas are lower in the case of the group of children and adolescents with DLD compared to the group with TD. In addition, taking into account that all the morphology subtests are assessed based on a maximum of 10 points, it is observed that in the group of people with DLD, the formation of plurals (both those ending in consonants and vowels), adjectives and the present tense in regular verb forms are the highest performing subareas, scoring above 5 points. However, in the group of people with TD, all the mean scores for the different subareas are above 5 points. The results indicate that there are significant differences (*p* < 0.01) in all subareas between both groups.

Regarding the gross total scores, the DLD group obtained an average of 78.53 points out of 190 points (39.4). The TD group has a score of 150.02 points out of 190 (SD = 35.2). This also translates when setting the percentile score, which shows the inequality between both groups. Differences between groups can be observed in Table 3.

Therefore, there are significant differences (*p* < 0.01) in the performance of tasks in the area of morphology between children and adolescents with DLD and children and adolescents with TD.

## 4. Discussion

The results of this study show how morphology is one of the weaknesses of people with DLD [44], in this case, in Spanish [46]. The data obtained follow the line of other research highlighting that grammatical morphology performance is lower in people with DLD compared to people with TD, both groups being of the same chronological age [22,25,31,47,48,49].

Within grammatical morphology, verbal morphology tends to present serious difficulties [50]. Kornilov, Grigorenko and Rakhlin [51] establish as an example that children with DLD may make a mistake when saying “She walks to school yesterday” instead of “She walked to school yesterday”. In English, this also occurs in the use of progressive verb tenses—ing [52]—and the use of these morphemes can serve as a clinical marker of DLD [25]. Our study has shown that, in this case, the use of verb tenses is seriously compromised, both in all the different verb forms and in the type of verb used, which would support the idea of a possible clinical differentiation with respect to people with TD. In the case of bilingual children, verbal morphology has been established as a possible clinical marker for the presence of DLD [53].

Similar to our study, Castilla-Earls, Auza, Pérez-Leroux, Fulcher-Rood and Barr [54] aimed to identify the morphological markers allowing the identification of people with DLD in Spanish-speaking children in Mexico compared to children with TD. To this effect, they were tasked with producing morphological strategies such as articles, direct object pronouns, adjectives, plurals, verb conjugations and the subjunctive in Spanish. The results showed that there were differences between both groups in all areas except in the formation of the plural. Furthermore, they concluded that pronouns and verbs should be considered morphological markers of DLD in monolingual Spanish-speaking children. Our results have shown that the differences are indeed significant in all areas of morphology, including the formation of plurals, both in the case of singulars ending in a vowel, and in the case of those ending in a consonant. This differentiation in terms of inflectional morphology of number has also been ascertained in the English language [50].

These specific grammatical morphological deficits in DLD could be explained by the Extended Optional Infinitive hypothesis (EOI) [23]. The EOI hypothesis predicts that morphemes belonging to closed categories are subject to an omission that may or may not appear. However, the use of morphemes in open categories should not be affected (for example, in the case of nominal and adjectival inflections). In this case, the morphological development of the child with DLD will present a considerable delay and will be more persistent over time [55]. This hypothesis has been accepted in other language alterations where morphology is altered, as in the case of Down syndrome [56,57]. In the case of DLD, there is no clear consensus on this hypothesis in terms of the language used.

Furthermore, an attempt has also been made to explain these errors through the Surface Hypothesis [29]. This hypothesis maintains that the greatest difficulties appear in those languages with a much more varied morphology. The frequent appearance of different morphemes makes their acquisition and consolidation much more difficult, so that children with DLD would be much more limited in their use. Some research has supported this hypothesis, especially in the case of verbal morphology in bilingual children [58].

Regarding the development of these characteristics, alterations in the area of morphology are evident throughout schooling. In this sense, it appears that one of the most typical error patterns in morphology in DLD seems to reside in the omission of certain morphemes more frequently and for a longer period of time than children with TD [22,25,59,60,61]. However, it must be taken into account that, in our study, the differences in morphology are evident from 6 years of age and may not be so evident at younger ages. For example, Thordardottir and Namazi [62] found that in French-speaking preschool-age children with DLD, errors in grammatical morphology were very infrequent and followed a developmental pattern very similar to that of children without DLD and matched in mean length of utterance. These researchers, however, highlight the differences with regard to other studies carried out in the English language that establish that verbs can be considered a clinical marker at preschool age in children with DLD [51,63].

As possible limitations of the study, we can highlight that the sample could have been larger, and the results should focus primarily on the Spanish language, but we cannot affirm that these characteristics are the same in other languages. Lastly, it is important to acknowledge that, although our results have shown additional evidence regarding the processing of morphology in DLD, the profound nature of these alterations remains to be determined. The strong evidence with this study is that morphology is an area in which special attention should be paid not only in order to analyse the morphological pattern and the most frequent types of errors in DLD, but also focusing on a much more direct component, such as intervention. So, speech-language therapy should incorporate intervention strategies that target specific morphological patterns and errors identified in this study and consider the specific difficulties in the area of morphology as an area of special interest during language intervention. In this manner, as a result of the results obtained, it would be advisable to integrate this work into speech therapy intervention in the group of people with DLD in order to minimise the oral language difficulties that the disorder itself involves.

## Figures and Tables

**Table 1 children-09-00125-t001:** Characteristics of the sample.

	Group with DLD	Group with TD
Mean chronological age	9.3 (SD = 2.9)	9.21 (SD = 2.94)
Range age (years)	5.66–14.75	5.83–14.75
Gender	18 females and 29 males	25 females and 18 males

**Table 2 children-09-00125-t002:** Score averages in the morphology subareas.

Morphology subareas	Group with DLD (*n* = 47)	Group with TD (*n* = 43)
Plurals: singulars ending in vowel	8.26 (2.2)	9.53 (0.9)
Plurals: singulars ending in consonant	7.38 (2.9)	9.58 (0.9)
Adjectives	8.21 (2.2)	9.70 (0.7)
Regular verb forms: Present	5.49 (3.2)	8.98 (2.3)
Regular verb forms: Past	4.66 (3.8)	7.86 (3.0)
Regular verb forms: Future	3.28 (3.7)	7.81 (3.4)
Regular verb forms: Imperfect	2.40 (3.0)	7.35 (3.4)
Irregular verb forms: Present	4.06 (3.2)	8.05 (2.7)
Irregular verb forms: Past	2.06 (2.5)	6.16 (2.9)
Irregular verb forms: Future	2.04 (3.1)	6.86 (3.6)
Participles	4.23 (3.2)	8.35 (2.3)
Comparatives and superlatives	3.43 (2.9)	7.33 (3.2)
Derived nouns: Professions	4.79 (2.7)	8.49 (1.4)
Derived nouns	2.57 (2.6)	7.4 (2.4)
Derived adjectives	3.32 (3.0)	7.65 (2.3)
Personal pronouns: Subjects	3.98 (3.5)	8.35 (2.1)
Personal pronouns acting as object	1.57 (2.8)	5.53 (3.7)
Reflexives	3.13 (3.0)	7.42 (3.0)
Possessives	3.66 (3.5)	7.63 (2.8)

Standard deviations in parentheses. All comparisons are statistically significant (*p* < 0.01).

**Table 3 children-09-00125-t003:** Scores in morphology area.

	Group with DLD (*n* = 47)	Group with TD (*n* = 43)
Total scores	78.53 (39.4)	150.02 (35.2)
Range total scores	44–190	12–170
Percentile scores	14.15 (23.8)	78.91 (25.8)
Range percentile scores	23–100	0–85
Transformed scores	39.62 (26.6)	78.43 (30.7)
Range transformed scores	23–100	6–93

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
