# Peer review of "Morphological Difficulties in People with Developmental Language Disorder"

_children, 2022, doi:10.3390/children9020125_

Round 1
Reviewer 1 Report
Reviewer Comments
Writing Issues:
- Abstract - “lies in the use of” is unnecessarily wordy.
- Abstract - “had Typical Development (TD)” is awkward; perhaps 47 were in the diagnosed DLD group, and 43 were in the TD group?
- Abstract - comparing “people with DLD” to “TD” is not parallel. I recommend referring to them as groups throughout the paper (e.g., the DLD group and the TD group).
- Abstract – The final sentence is not a clear conclusion: These data have led us to explore explanations based on several hypotheses and to confirm the research outlining the explicit morphological difficulties in people with DLD in order to establish a clinical diagnosis and a subsequent intervention. Shorten the statement to make sure it says what you mean.
- Line 25 - hereinafter should be hereafter
- Lines 26-28 – Use parallel structure for clarity: deficits in morphosyntactic constructions, lexical reduction and alterations in speech, social participation, communication and academic performance (i.e., you are not talking about deficits in alterations in speech).
- Line 62 – lesser should be less
- Line 37 – y should be and
- Line 102 – both groups should be the groups
- Participants – again, use parallel language for the groups; I recommend the DLD group and the TD group
- Participants – use female and male, instead of women and men, since your participants are children
- Line 93 / 217 – avoid proven. Perhaps likely, or evidence strongly suggests that…
Literature Review:
- Lines 84-87 – Explain the relevance of this point to your study: Difficulties in morphology not only emerge when comparing children with DLD 84 with others of the same chronological age, but also appear with children with Typical 85 Development (TD) who are younger in age, but matched in mean length of utterance 86 (MLU) [27-32, 22].
Methods:
- Line 107 – change to: In this research, the morphology section of the Objective and Criterial…
- Line 114 – instead of this area of the test, name the section you used.
Discussion:
- Line 161 – especially in Spanish seems to contradict with information in your literature review about English morphemes being particularly difficult for people with DLD. Clarify in one or both places.
- Lines 169-170 – Does this line imply that morphology is already being used as a clinical marker for DLD?: the use of these morphemes can serve as a clinical marker of 169 DLD [25]. If not, rephrased the sentence to that clear. If so, incorporate this into your literature review since, at present, the lit review reads as though morphology is not yet being used a clinical marker. You make a similar claim in lines 172-174, but use the phrasing possible clinical marker. Be clear with what has and hasn’t been established, and make sure what you say in your Discussion is consistent with your lit review.
- The information about the Castilla-Earls, Auza, Pérez-Leroux, Fulcher-Rood & Barr [53] study belongs in the lit review. New study findings should not be introduced in the discussion.
- Lines 197-198 – Explain how this relates to your claims about morphological difficulties in English and Spanish: This hypothesis maintains that the greatest difficulties appear in 197 those languages with a much more varied morphology.
- The discussion section is not clearly organized. Offer an introductory paragraph that outlines what you will address in the discussion: potential theoretical explanations, implications for practice, etc. Also, the stated goal of the paper was this: The goal of this research was to discover the distinctive 11 characteristics of the different morphological subareas in people with DLD. Therefore, the findings and discussion section should address these much more thoroughly, exploring the results in each of the 11 categories in more depth and the implications for using morphological assessment as a reliable diagnostic for DLD.
Author Response
Dear reviewer:
Thank you very much for your comments. We have changed some information and we have followed your recommendations.
Writing Issues:
Abstract - “lies in the use of” is unnecessarily wordy. CHANGED
Abstract - “had Typical Development (TD)” is awkward; perhaps 47 were in the diagnosed DLD group, and 43 were in the TD group? CHANGED
Abstract - comparing “people with DLD” to “TD” is not parallel. I recommend referring to them as groups throughout the paper (e.g., the DLD group and the TD group). CHANGED
Abstract – The final sentence is not a clear conclusion: These data have led us to explore explanations based on several hypotheses and to confirm the research outlining the explicit morphological difficulties in people with DLD in order to establish a clinical diagnosis and a subsequent intervention. Shorten the statement to make sure it says what you mean. CHANGED
Line 25 - hereinafter should be hereafter. CHANGED
Lines 26-28 – Use parallel structure for clarity: deficits in morphosyntactic constructions, lexical reduction and alterations in speech, social participation, communication and academic performance (i.e., you are not talking about deficits in alterations in speech).
We are talling about basic lingüistic characteristics in language and speech
Line 62 – lesser should be less. CHANGED
Line 37 – y should be and. CHANGED
Line 102 – both groups should be the groups. CHANGED
Participants – again, use parallel language for the groups; I recommend the DLD group and the TD group. CHANGED
Participants – use female and male, instead of women and men, since your participants are children. CHANGED
Line 93 / 217 – avoid proven. Perhaps likely, or evidence strongly suggests that…CHANGED
Literature Review:
Lines 84-87 – Explain the relevance of this point to your study: Difficulties in morphology not only emerge when comparing children with DLD 84 with others of the same chronological age, but also appear with children with Typical 85 Development (TD) who are younger in age, but matched in mean length of utterance 86 (MLU) [27-32, 22]. A new sentence has been added.
Methods:
Line 107 – change to: In this research, the morphology section of the Objective and Criterial…CHANGED
Line 114 – instead of this area of the test, name the section you used. CHANGED
Discussion:
Line 161 – especially in Spanish seems to contradict with information in your literature review about English morphemes being particularly difficult for people with DLD. Clarify in one or both places. CHANGED
Lines 169-170 – Does this line imply that morphology is already being used as a clinical marker for DLD?: the use of these morphemes can serve as a clinical marker of 169 DLD [25]. If not, rephrased the sentence to that clear. If so, incorporate this into your literature review since, at present, the lit review reads as though morphology is not yet being used a clinical marker. You make a similar claim in lines 172-174, but use the phrasing possible clinical marker. Be clear with what has and hasn’t been established, and make sure what you say in your Discussion is consistent with your lit review.
The information about the Castilla-Earls, Auza, Pérez-Leroux, Fulcher-Rood & Barr [53] study belongs in the lit review. New study findings should not be introduced in the discussion.
A new Pharagraph has been added in the introduction part about morphology as a possible clinical marker.
Lines 197-198 – Explain how this relates to your claims about morphological difficulties in English and Spanish: This hypothesis maintains that the greatest difficulties appear in 197 those languages with a much more varied morphology.
The discussion section is not clearly organized. Offer an introductory paragraph that outlines what you will address in the discussion: potential theoretical explanations, implications for practice, etc. Also, the stated goal of the paper was this: The goal of this research was to discover the distinctive 11 characteristics of the different morphological subareas in people with DLD. Therefore, the findings and discussion section should address these much more thoroughly, exploring the results in each of the 11 categories in more depth and the implications for using morphological assessment as a reliable diagnostic for DLD.
That has been included in the last paragraph
Reviewer 2 Report
Give a summary of the findings and discuss the most salient outcomes in light of the theme and literature.
Conclusions are related to the empirical and theoretical parts of the article and this part is missing now. All research questions should receive a comprehensive and reasoned answer. Conclusions must show why your article is novel and open new research perspectives. Conclusions are deliberative, they include your assessment of previous research and your own methodological choices.
Please clearly outline recommendations and further research opportunities, as well as the limitations and contributions of your article.
Author Response
Dear reviewer:
Thank you for your comments. We have follow your recommendations and we have added new information in the last paragraph.
Best regards